# The Effects of the Dietary and Nutrient Intake on Gynecologic Cancers

**DOI:** 10.3390/healthcare7030088

**Published:** 2019-07-07

**Authors:** Masafumi Koshiyama

**Affiliations:** Department of Women’s Health, Graduate School of Human Nursing, The University of Shiga Prefecture, Shiga 522-8533, Japan; koshiyamam@nifty.com; Tel.: +81-749-28-8664; Fax: +81-749-28-9532

**Keywords:** diet, nutrition, cervical cancer, endometrial cancer, ovarian cancer

## Abstract

The contribution of diet to cancer risk has been considered to be higher in advanced countries than in developing countries. In this paper, I review the current issues (a review of the relevant literature), and the effects of the dietary and nutrient intake on three types of gynecologic cancer (cervical, endometrial and ovarian cancers). In cervical cancer, the most important roles of diet/nutrition in relation to cancer are prophylaxis and countermeasures against human papillomavirus (HPV) infection. The main preventive and reductive factors of cervical cancer are antioxidants, such as vitamin A, C, D and E, carotenoids, vegetables and fruits. These antioxidants may have different abilities to intervene in the natural history of diseases associated with HPV infection. For endometrial cancer, the increase in peripheral estrogens as a result of the aromatization of androgens to estrogens in adipose tissue in obese women and insulin resistance are risk factors. Thus, we must mainly take care to avoid the continuous intake of fat energy and sugar. In ovarian cancer, the etiology has not been fully understood. To the best of our knowledge, the long-term consumption of pro-inflammatory foods, including saturated fat, carbohydrates and animal proteins is a risk factor. The intake of acrylamide is also a risk factor for both endometrial and ovarian cancer. Most papers have been epidemiological studies. Thus, further research using in vitro and in vivo approaches is needed to clarify the effects of the dietary and nutrient intake in detail.

## 1. Introduction

As a source of important physiologically functional components of human beings, diet and nutrition play a vital role in the management of cancer. Dietary and nutritional factors have been estimated to contribute to 20–60% of cancers worldwide [1,2]. Among them, the contribution of diet to the cancer risk in developed countries has been considered to be relatively higher [3,4,5], whereas that in developing countries has been considered to be lower, perhaps around 20% [6,7]. Thus, cancer may be considered to be a disease of developed countries. Understanding the effects of diet and nutrition on cancer risk is very important for public health. For these reasons, it is necessary to review the dietary and nutrient intake in developed countries.

This paper reviews the current issues (a review of the relevant literature), and the effects of the dietary and nutrient intake on three types of gynecologic cancer (cervical, endometrial and ovarian cancers) and discusses the roles of the dietary and nutrient intake in relation to each type of cancer. 

## 2. Cervical Cancer

Persistent infection by high-risk human papillomavirus ([HPV]-type 16, 18, 31, 33, 35, 39, 45, 51, 52, 56, 58, 59 and 68) genotypes has been recognized as a necessary step for the development, maintenance and progression of cervical intraepithelial neoplasia (CIN) and cervical cancer. However, up to 60% of pre-invasive intraepithelial lesions regress spontaneously with progression to invasive squamous carcinoma (ISC) a relatively uncommon event [8]. Some cofactors that influence the risk of progression from HPV infection to persistent HPV and the development of squamous intraepithelial lesions (SIL) are needed [9]. These include environmental, immunologic and lifestyle cofactors such as cigarette smoking, diet, long-term oral contraceptive (OC) use, high parity and coinfection with other sexually transmitted infections [10]. In particular, cigarette smoking plays a role in the early stages of carcinogenesis, from the acquisition of HPV infection to the development cervical cancer; the risk of cervical cancer in smokers is two-fold that of non-smokers [11]. Vaccination and safe sex education have established as effective prevention strategies against HPV infection, CIN and ISC [12].

During recent decades, the important role of antioxidant vitamins in preventing the development of cervical carcinogenesis has received much attention [13]. Antioxidant vitamins can act as efficient scavengers of free radicals and oxidants to prevent free-radical damage to DNA [14]. Moreover, if the free radicals and oxidants are not neutralized by antioxidant molecules, the inflammatory processes caused by HPV infection could lead to extensive damage to DNA proteins [15]. Vitamins such as vitamin A (retinoic acid), C (ascorbic acid) and E (tocopherol) can also inhibit the proliferation of cancer cells [9,16], stabilize the p53 protein [17], prevent DNA damage, reduce immunosuppression [18,19], and support the receptor signal transduction pathways [19].

Cervical cancer develops through HPV infection, CIN1, CIN2 and CIN3. To provide an understanding of the preventive and complementary effects, the relationship between the effects of the dietary and nutrient intake, and the development of cervical cancer (HPV infection, CIN1, CIN2, CIN3 and cervical cancer) are now described.

### 2.1. HPV Infection and Nutrients

Recently, Barchitta et al. reported that a Western diet, which is characterized by the high intake of red and processed meats, dipping sauces, chips, and snacks with a low intake of olive oil, was associated with a higher risk of HPV infection [20]. A Western diet has been reported to lead to increased inflammation, reduced infection control and increased risk of developing auto-inflammatory disease [21]. In comparison to women who showed low adherence to a Mediterranean diet (MD), which includes vegetables legumes, fruits and nuts, cereals, fish, and a high ratio of unsaturated to saturated lipids, women with medium adherence to an MD had a lower risk of HPV infection [20]. Meanwhile, the low consumption of whole vegetables (including dark green, dark yellow, and dark orange vegetables and cruciferous vegetables), fruits (including fruit juices), yogurt, tofu, fish and meat was not protective against HPV persistence [22]. Supporting these results, Sedjo et al. reported that higher levels of vegetable consumption were associated with a 54% reduction in the risk of HPV persistence (odds ratio [OR]: 0.46) in a prospective cohort study [23].

Giuliano et al. demonstrated that the dietary intake of carotenoids (lutein, zeaxanthin, β-cryptoxanthin), vitamin-C and papaya was associated with a reduced incidence of type-specific HPV persistence in 433 HPV-positive women [24]. The intake of carotenoid (lutein) was also reported to be associated with a 50%–63% reduction in the risk of HPV persistence in a prospective cohort study [23].

### 2.2. CIN 1 and Nutrients

In a case-control study of 265 HPV-positive women, higher consumption of papaya also reduced the SIL risk [25]. Moreover, the nutrient intake of carotenoids (β-cryptoxanthin and α-carotene) was also marginally protective against SIL. These data are consistent with data from the report of Giuliano et al. [24], which showed a reduced incidence of type-specific HPV persistence. 

Vahedpoor et al. demonstrated that 29 patients with CIN1 who took one 50,000 IU dose of vitamin D supplement every 2 weeks for 6 months showed a higher rate of reduced insulin metabolism marker levels and improved levels of biomarkers of inflammation and oxidative stress such as nitric oxide (NO), total antioxidant capacity (TAC), total glutathione (GSH) and malondialdehyde (MDA), in comparison to 29 placebo controls [26]. Thus, they hypothesized that the anti-inflammatory and anti-oxidative effects of vitamin D actions may be useful for reducing the risk of CIN 1. The report of Schulte-Uebbing et al. which showed that treatment with vitamin D vaginal suppositories (12,500 IU, three nights a week, for 6 weeks) resulted in antidysplastic effects in patients with CIN 1, but not those with CIN 2, supports this hypothesis [27].

Vitamin A (retinol) is needed for the replication of basal mucosal-cells and the synthesis of protein blocks [18]. Thus, vitamin A deficiency may be associated with an increased risk of developing squamous metaplasia and HPV infection. Yeo. et al. showed that low serum retinol levels were associated with an increased risk of developing CIN1 [28]. Vitamin A (retinol), vitamin D and papaya may be more effective in preventing low-grade CIN. 

### 2.3. CIN 2 and Nutrients

Feng et al. showed that the consumption of onion vegetables, legumes and nuts was associated with adjusted odds ratios (AORs) for the development of CIN2+ were 0.589, 0.591 and 0.635, respectively, among 748 HPV-positive women [29]. The consumption of these foods may reduce the risk of developing CIN2+. However, no associations were found between the risk of developing CIN2+ and the consumption of dark-colored vegetables, light-colored vegetables or eggs. 

Women with a lower intake of vegetables and fruits, and a higher viral load (≥15.5) have a higher risk of developing CIN2/3 in comparison to women with a lower intake of vegetables and fruits, and a lower viral load (<15.5) [30]. Thus, high-risk patients, especially those infected by high-risk HPV types, should increase and maintain their intake of fruits and vegetables at moderate levels [22]. Similarly, green tea (OR, 0.551) and the intake of vegetable (OR, 0.896) were identified as protective factors against CIN2/3 [31]. Among 328 HPV-positive women (control, *n* = 166; CIN1, *n* = 90; and CIN2/3, *n* = 72), patients who took multivitamins and had a lower HPV viral load (<15.5) had a significantly lower incidence of CIN1 (OR, 0.35) and CIN2/3 (OR, 0.11), and the use of dietary supplements, including multivitamins (OR, 0.21), vitamin A (not retinol) (OR, 0.19), vitamin C (OR, 0.24), vitamin E (OR, 0.20), and calcium (OR, 0.21) was associated with a low risk of developing CIN2/3 [32]. With regard to CIN3, however, calcium intake had no apparent protective effect in a Japanese case-control study [33]. As described above, these antioxidants may have different abilities to intervene in the natural history of diseases associated with HPV infection. 

In a case-control study, Kwanbunjan et al. showed that low serum folate levels were associated with a high-risk of developing both CIN1 and CIN2/3 [34]; however, medium serum folate levels were associated with a high-risk of developing CIN1 but not CIN2/3. Thus, dose-responsive serum folate levels might be useful for CIN2/3 prevention [35]. Folate has important roles in DNA synthesis, repair, methylation and cell proliferation [36]. 

### 2.4. CIN 3 and Nutrients

Tomita et al. suggested that there is an additive interaction between a low intake of fruits and vegetables, especially fruits and dark-green and deep-yellow vegetables, smoking habits and the risk of developing CIN3 [37]. In addition, low serum levels of lycopene, vitamin A (retinol) and vitamin E (alpha tocopherol and gamma tocopherol) tended to increase the risk of developing CIN3, whereas medium to high levels of serum carotenoids (alpha-carotene, beta-cryptoxanthin, lutein/zeaxanthin), vitamin E (gamma tocopherol) and particularly serum lycopene could reduce the risk of CIN3 development [22,37]. From these findings, some antioxidants might be effective in preventing the development of CIN3 if the administration was initiated from the stage of cervical inflammation (which promotes HPV infections), rather than when the patient is directly infected by HPV [38]. In brief, vitamin E, folate and lycopene might be more effective for preventing high-grade CIN rather than primary HPV infection.

### 2.5. Cervical Cancer and Nutrients

A meta-analysis of studies assessed that the intake of carotene and carotenoid strongly reduced the cervical cancer risk, whereas the intake of vitamin A (retinol) weakly reduced it [39]. As described above, a lower vitamin A (retinol) intake might be associated with early events in the HPV infection process [28]. In another meta-analysis, carotene levels were also inversely associated with the invasive cervical cancer risk (OR, 0.68) [40]. On the other hand, retinol had no preventive effect on cervical cancer. Zhang et al. also indicated that inverse associations between serum carotenoid (α-carotene, β-carotene, and lutein/zeaxanthin) and tocopherol (α-tocopherol) concentrations and the risk of cervical cancer, but a null association for retinol [41]. 

In a prospective study of 299,649 women, participating in the European Prospective Investigation into Cancer and Nutrition study, a statistically significant inverse association of ISC with a 100 g increase in the daily total fruit intake (hazard ratio [HR] 0.83) and a statistically nonsignificant inverse association with a 100 g increase in the daily total vegetable intake (HR 0.85) was observed [42]. Their findings showed that the effects of fruit might play a role in invasive cervical cancer but not in CIN, thus suggesting that if there was any true effect, it would occur in a late stage of the cancer process (CIN1 < CIN2/3 < ISC). Another case-control study supported an inverse association between ISC and fresh fruit intake [43].

Some experimental studies have investigated the relationship between dietary intake and cervical cancer. Recently, Yang et al. reported that dietary oleic acid exerted a stimulatory effect on cervical cancer growth and metastasis [44]. They suggested that oleic acid-induced CD36 might be a promising therapeutic target that acts against cervical cancer through an Src/ERK-dependent signaling pathway. In cervical cancer cell lines, Cheng et al. investigated the effects of sulforaphane which is a natural, compound-based drug derived from dietary isothiocyanates, on anti-proliferation and G2/M phase cell cycle arrest [45]. They suggested that sulforaphane might delay the development of cervical cancer cells by arresting cell growth in the G2/M phase via the down-regulation of the cyclin B1 gene expression, dissociation of the cyclin B1/CDC2 complex, and the up-regulation of GADD45β proteins. Using cervical cancer cell cultures, Alshatwi et al. reported the effect of tea polyphenols (TPPs) [46]. They demonstrated that TPPs enhanced the therapeutic properties of bleomycin (BLM). Moreover, TPP-BLM synergistically inhibited uterine cervical cancer cell viability by decreasing proliferation through apoptosis. They suggested that in combination with chemotherapy TPP might be an effective agent for treating uterine cervical cancer. Singh et al. reported that green tea compounds such as epigallocatechin gallate (EGCG: one of the polyphenols) are capable of chemosensitizing cervical cells (HeLa, SiHa) to cisplatin through the enhancement of cytotoxicity and by the induction of apoptosis due to the generation of excessive reactive oxygen species (ROS) [47]. Jakubowicz-Gil et al. reported that quercetin (one of the flavonoids) helps HeLa cells to become more sensitized to the apoptosis caused by cisplatin [48]. Xu et al. reviewed apigenin (one of the flavonoids) and its effects on cervical cancer cells and revealed that this compound can sensitize HeLa cells to paclitaxel-induced apoptosis through the enhancement of the intracellular accumulation of ROS [49].

On the other hand, Lin et al. showed a radiosensitizing enhancement ratio of 1.65 with the combination of radiotherapy and quercetin (one of the flavonoids) [50]. Shin et al. also reported that genistein (one of the flavonoids) behaves as a radiosensitizer in a cervical cancer cell line (CaSki) leading to the induction of apoptosis via the modulation of ROS and a decrease in cellular viability due to the downregulation of the expression of E6 and E7 [51]. In a recent randomized double-blind clinical trial, Wuryanti et al. showed that dietary supplementation enriched with polyunsaturated fatty acid (PUFA) can reduce inflammation in patients with advanced cervical cancer, and that it may improve the cancer cell response to radiation because the reduced serum prostaglandin E2 (PGE2) level would reduce the survival of cancer cells [52]. 

Table 1 summarizes the preventive and reductive effects of the dietary/nutritional intake on HPV infection, CIN 1, CIN 2, CIN 3 and cervical cancer. In Table 1, the main preventive and reductive factors are antioxidants, such as vitamin A, C, D and E, carotenoids, vegetables and fruits. These antioxidants may have different intervention effects on the natural history of diseases associated with HPV infection. Vitamin A (retinol), vitamin D and papaya might be more effective in preventing low-grade CIN. In contrast, vitamin E and lycopene might be more effective in preventing high-grade CIN. Both fruits and carotenoids might be widely effective in preventing HPV infection, CIN and cervical cancer. Vegetables might be effective in preventing HPV infection and CIN except for cervical cancer.

## 3. Endometrial Cancer

Endometrial cancers have been classified into Type I (endometrioid) and Type II (nonendometrioid) cancers [53]. Type I cancers, which account for most endometrial carcinomas, occur in association with endometrial hyperplasia in perimenopausal women. Unopposed estrogen stimulation is thought to play an important role in this type of carcinogenesis. In contrast, Type II cancers may develop de novo in the atrophic endometrium of older postmenopausal women. Obesity and estrogen replacement therapy are well-defined risk factors for Type I cancers, with reported relative risks ranging from 2 to 10 [54]. The risk can be explained by the increased availability of peripheral estrogens as a result of the aromatization of androgens to estrogens in adipose tissue and lower concentrations of sex hormone-binding globulins in obese women. Hyperinsulinemia and diabetes have also been independently associated with increased endometrial cancer risk [55]. Insulin can stimulate the growth of endometrial cells by binding to insulin receptors in the endometrium [56] and by increasing free insulin-like growth factor-1 levels via a reduction in the levels of insulin-like growth factor-binding protein-1 [57]. 

In the case of endometrial cancer, the direct etiology has not been clearly understood. To provide an understanding of high-risk diets and the preventive effects of diet, the relationship between dietary intake and the endometrial cancer risk is discussed.

### 3.1. Energy 

Chu et al. performed a meta-analysis of 17 studies to investigate the effect of the overall energy intake on the endometrial cancer risk [58]. They found that there was no association between the total energy intake and the endometrial cancer risk in either overall group (OR = 1.11). In a specific macronutrient calorie analysis, a higher fat energy intake was found to be associated with an increased endometrial cancer risk (OR = 1.72), while energy from the intake of carbohydrates and proteins was not. The incidence of endometrial precancerous lesions was also found to be higher in overweight and obese patients [59]. On the other hand, a meta-analysis showed that physical activity reduced the endometrial cancer risk [60].

### 3.2. Sugar 

In a prospective cohort study of postmenopausal women, Inoue-Choi et al. reported that a higher intake of sugar-sweetened beverages (SSBs) may increase the type I endometrial cancer risk, regardless of body weight, physical activity, history of diabetes, and cigarette smoking [61]. They suggested that the association between a higher intake of SSBs and a higher type I cancer risk became slightly stronger after adjustment for the body mass index (BMI). Factors other than body weight (e.g., an overall unhealthy lifestyle) may be involved in the link between intake of SSBs and type I cancer. In another report, women in the highest quartile of added sugar intake also had a significantly increased endometrial cancer risk (OR = 1.84) [62]. Among women with a waist–to–hip ratio (WHR; a marker of central obesity) of ≥0.85, the endometrial cancer risk of the women in the highest tertile of added sugar intake was significantly higher than that of the women in the lowest tertile of added sugar intake (OR = 2.50). 

### 3.3. Meat, Fish and Fatty Acid 

In a case control study that included 454 Italian women with endometrial cancer and 908 controls, the intake of “animal-derived nutrients and polyunsaturated fatty acids (PUFAs)” was associated with a significantly higher endometrial cancer risk among obese women [63]. They concluded that the recommendation of reducing the high intake of products of animal origin has the potential to reduce the endometrial cancer risk. On the other hand, long-chain ω-3 (n−3) PUFAs, such as eicosapentaenoic acid (EPA; 20:5ω-3) and docosahexaenoic acid (DHA; 22:6ω-3), which are derived from fish, are thought to be anti-inflammatory; however, some studies of fish consumption suggested that it was associated with an increased endometrial cancer risk [64]. In a prospective cohort study that included 22,494 normal women and 263 women with endometrial cancer, it was found that the intake of long-chain ω-3 PUFAs and their food sources increased the endometrial cancer risk among overweight and obese women. Similarly, a pooled analysis of seven cohort studies and 14 case-control studies suggested that the endometrial cancer risk was significantly increased by 5% per 10% kilocalories from the total fat intake and by 17% per 10 g/1000 kcal of the saturated fat intake [65]. 

Adiponectin is an adipokine that was shown to be a risk factor for endometrial cancer [66]. Shang et al. built a rat model with insulin resistance (IR) and endometrial hyperplasia (EH) and analyzed morphological effects induced in the rat uterus after high-fat diet (HFD) feeding and estrogen treatment [67]. The uterine adiponectin gene and protein levels in the HFD-estradiol group were higher than those in the HFD group. Thus, they might mainly represent a response to estradiol; the expression of which was increased in rats with EH in comparison to rats without EH.

### 3.4. Dietary Acrylamide 

Acrylamide was evaluated by the International Agency for cancer (IARC) as a “probable human carcinogen (class 2A)” in 1994, based on positive evidence from animal studies and inadequate epidemiological evidence [68]. Acrylamide is also found in common human foods (e.g., potato crisps, fried potato, French fries, cookies, coffee, fried almonds, etc.) that are mainly formed during high-temperature cooking as part of the Maillard browning reaction [69]. In animal experiments, positive dose–response relationships between acrylamide administered in drinking water and several types of cancers were shown, especially in hormone-sensitive organs, including the uterus [70]. In four large prospective cohort studies that included 453,355 women and 2,019 women with endometrial cancer, no association was found between the dietary acrylamide intake and the overall endometrial cancer risk (pooled relative risk [RR] for high vs. low intake = 1.10) [71]. However, a high intake of acrylamide was significantly associated with an increased endometrial cancer risk among female never-smokers (pooled RR for high vs. low intake = 1.39). In a similar way, Obon-Santacana et al. also concluded that the dietary intake of acrylamide was not associated with the overall cancer risk or type I endometrial cancer risk; however, a positive association with type I endometrial cancer was observed in women who were both non-users of oral contraceptives and never-smokers [72]. Hogervorst et al. analyzed acrylamide-gene interactions associated with the endometrial cancer risk in the prospective Netherlands Cohort Study on diet and endometrial cancer, which included 62,573 women [73]. They then adjusted for multiple comparisons. Their study revealed that there was no statistically significant interaction between single nucleotide polymorphisms (SNPs) and acrylamide intake for endometrial cancer risk. However, they found nominally statistically significant interactions with SNPs in acrylamide-metabolizing enzymes; CYP2E1(rs915906 and rs2480258) and deletions of GSTM1 and GSTT1. It is generally thought that acrylamide may cause cancer through the genotoxic action of acrylamide’s metabolite glycidamide (generated by the action of cytochrome P4502E1(CYP2E1)). 

### 3.5. Alcohol 

The consumption of alcohol may increase the endometrial cancer risk by adversely affecting the concentrations of sex steroids in both premenopausal and postmenopausal women [74,75,76]. In a recent, large (301,051 subjects), multicenter and prospective study, however, it was reported that alcohol consumption was not associated with the endometrial cancer risk [77]. The findings did not contradict those of previous cohort studies that reported either no association [78,79] or a non-significant increase in the endometrial cancer risk with the highest category of alcohol intake [80,81]. Conversely, Liu et al. found that the consumption of less than one alcoholic drink per day had a significant protective effect against endometrial cancer, but that no significant association was observed in women who consumed more than one drink per day [82]. In recent large (68,067 women) prospective cohort study, Je et al. also reported that women with a light alcohol intake of <5 g of ethanol per day (approximately half a drink per day) was associated with a 22% reduction in the endometrial cancer risk in comparison to non-drinkers, but that a higher intakes did not provide any additional benefits [83]. They also found that a light alcohol intake was associated with a low endometrial cancer risk among obese women. They uniquely speculated that obese women tend to have insulin resistance and higher levels of insulin in comparison to lean women and that potential ability of alcohol to improve those conditions may have led to a reduced endometrial cancer risk. On the other hand, lean women who tend to have low insulin resistance and normal fasting insulin levels may be more likely to be affected by the increase in estrogens that results from the intake of alcohol.

### 3.6. Dietary Carbohydrate 

Most studies have reported that non-significant elevations in the endometrial cancer risk according to the dietary glycemic load (GL = both carbohydrate quality and quantity) or glycemic index (GI = carbohydrate quality) [84]. Recently, Brenner also reported that the intake of foods eliciting a glycemic response was not associated with the endometrial cancer risk in Canadian women (511 cases and 980 controls) [85]. They suggested that the dietary carbohydrate content in the year prior to diagnosis may not have an important etiological role in endometrial cancer because their findings that a higher GL did not increase the endometrial cancer risk. On the other hand, in a prospective cohort study of prostate, lung, colorectal and ovarian cancer screening (PLCO), significant inverse associations were detected between the endometrial cancer risk and the total available carbohydrate intake ([HR] = 0.66), total sugar intake (HR = 0.71), and GL (HR = 0.63) when women in the highest quartiles of intake were compared to those in the lowest quartiles [86]. Thus, they insisted that the high intake of carbohydrates and a high GLs had a protective effect against the development of endometrial cancer.

Insulin resistance and metabolic syndrome are thought to be risk factors for endometrial cancer [87,88]. The fasting insulin level, a marker of insulin resistance, was reported to be associated with the overall endometrial cancer risk in a Canadian population-based case-control study [89]. Prescott et al. hypothesized that insulin-resistant individuals (obese or inactive) who consume an insulinogenic diet would be at the highest risk [90]. However, they did not observe significant associations between dietary insulin scores and the endometrial cancer risk among these individuals. They also concluded that dietary measures alone may not sufficiently capture absolute long-term insulin exposure, considering the complex interplay of diet, lifestyle and genetic factors, which contribute to hyperinsulinemia.

### 3.7. Flavonoid 

From an Italian case-control study including 454 women with endometrial cancer, the effects of flavonols, flavanones, flavonols, anthocyanidins, flavones, isoflavones and proanthocyanidins were examined [91]. As a result, a high dietary intake of proanthocyanidins with trimers (or polymers composed of more than three monomers) reduced the endometrial cancer risk, particularly in normal-weight women. No significant overall association was found for any other class of flavonoids. In cell culture experiments, kaempherol (a natural dietary flavonoid) successfully suppressed the viability of two estrogen receptor (ER)-positive endometrial cancer cell lines [92]. Kaempferol is a dietary bioflavonoid with anticancer, anti-inflammatory, and anti-oxidant properties that suppress cellular proliferation in human cancers through various mechanisms, including the induction of tumor suppressor p53 and the inhibition of ERα [93]. Isoflavones (one of the flavonoids) are structurally similar to 17β-estradiol. Thus, isoflavones possess selective estrogen receptor modulator (SERM)-like activity, with varying estrogenic and anti-estrogenic effects depending on the receptor characteristics of the target tissue [94]. Actually, three-year isoflavone soy protein (ISP) supplementation had no effect on the endometrial thickness or the rates of endometrial hyperplasia and cancer in postmenopausal women in a randomized, double-blind, placebo-controlled trial [95]. In a Japanese population-based prospective cohort study, there was also no evidence of a protective association between the intake of soy food or isoflavones, and the risk of endometrial cancer [96]. However, a meta-analysis of electric databases revealed that the consumption of larger amounts of dietary isoflavones from soy products and legumes weakly decreased the risk of endometrial cancer [97]. The authors hypothesized that the inconsistent conclusions likely resulted from differences in the characteristics (e.g., endogenous estradiol levels and BMI) of the study populations in different countries. Another report indicated that the intake of soy-containing foods was associated with a lower endometrial cancer risk [98].

## 4. Ovarian Cancer

Histopathological, molecular and genetic studies have recently provided a better model of ovarian carcinogenesis, including two broad categories, designated as type I carcinomas (low-grade serous, mucinous, endometrioid, clear cell and transitional cancers), in which precursor lesions in the ovaries have clearly been described, and type II carcinomas (high-grade serous cancers), in which such lesions have not been clearly described and in which tumors may develop de novo from the tubal and/or ovarian surface epithelium [99,100]. Type I carcinomas are, in general, slow-growing, indolent neoplasms that often develop from benign ovarian cysts. In contrast, type II carcinomas are high-grade clinically aggressive neoplasms, which are often found at an advanced stage. In particular, type II carcinoma is associated with a poor prognosis. 

There are some problems associated with screening to detect early-stage ovarian cancer [100,101]. Thus, every effort should be made to reduce its occurrence. 

The etiology of ovarian cancer is not understood. To provide an understanding of high-risk diets and the preventive effects of diet, the relationship between dietary intake and the ovarian cancer risk is discussed.

### 4.1. Pro-Inflammatory Food (Saturated fat, Carbohydrates, High-Glycemic Carbohydrates, etc.)

Chronic inflammation has been implicated as an underlying mechanism contributing to ovarian carcinogenesis [102]. Inflammation can influence tumor growth through the stimulation of DNA damage, the promotion of increased cell division that can give rise to DNA repair aberrations, the promotion of angiogenesis and the facilitation of invasion [102,103]. In addition, several factors that enhance inflammation have been associated with an increased ovarian cancer risk (e.g., application of body powder to external genitalia, endometriosis, pelvic inflammatory disease, and other factors) [104,105,106].

Cavicchia et al. developed a novel literature-derived tool to assess the inflammatory potential of a diet: the dietary inflammatory index (DII) [107]. The improved DII has been validated by examining its relationship to inflammatory biomarkers (e.g., CRP and interleukin-6) [108,109,110]. A pro-inflammatory diet is high in foods rich in saturated fat and carbohydrates, and low in foods rich in poly-unsaturated fatty acids, flavonoids and other dietary components, including a variety of vitamins and minerals [108]. A recent Italian case-control study revealed that the ovarian cancer risk was increased among Italian women who consumed a more pro-inflammatory diet [111]. Peres et al. also reported that a more pro-inflammatory diet was associated with an increased ovarian cancer risk (OR = 1.10), especially among African-American women older than 60 years of age [106]. In a more recent case-control study conducted in New Jersey (USA), most diets with pro-inflammatory DII scores also increased the ovarian cancer risk among post-menopausal women (OR for quartile 4 vs. quartile 1 = 1.89, *p* = 0.03) [112].

In the European Prospective Investigation into Cancer and Nutrition study, the ovarian cancer risk in individuals with a high intake of saturated fat was found to be higher than that in those with a low intake of saturated fat; however, there was no evidence of a dose-response relationship [113]. In the National Institutes of Health (NIH)-American Association of Retired Persons (AARP) Diet and Health Study, it was also reported that the intake of fat, particularly fat from animal sources, increased the ovarian cancer risk [114]. In the African American Cancer Epidemiology Study (AACES), the OR of the highest quartile of total carbohydrate intake vs. the lowest quartile of total carbohydrate intake was 1.57 (*p* = 0.03) [115]. Qiu et al. also indicated that the increased consumption of total fat, saturated fat and trans-fat might be associated with an increased ovarian cancer risk in a meta-analysis of 16 case-control and nine cohort studies [116]. Interestingly, they suggested that saturated fats can significantly increase the risk of developing serous and endometrioid ovarian cancer and that the serous ovarian cancer risk (Type II) was more susceptible to influence from the dietary fat intake. These findings support the hypothesis that the intake of pro-inflammatory foods can increase the ovarian cancer risk. 

In contrast, a higher intake of ω-3 fatty acid was reported to have protective effects against overall ovarian cancer, and endometrioid tumors in particular, in a New England case-control study that included 1872 cases and 1978 population-based controls [117]. It was indicated that polyunsaturated fatty acids, such as DHA inhibited proliferation in ovarian cancer cell lines via G1 cell cycle arrest and the induction of apoptosis and cellular stress [118].

### 4.2. Animal and Plant Protein 

Positive associations between a high intake of red meat [119,120] or processed meat [120,121] and the ovarian cancer risk were reported in case-control studies. Red meat and processed meat are sources of saturated fat and iron, which have been independently linked to carcinogenesis [122,123] as well as the fat content of the meat [123]. In addition, processed meats contribute to the formation of carcinogenic and mutagenic N-nitroso compounds [124] and heterocyclic amines [125]. 

In contrast, some case-control studies have suggested that the frequent intake of poultry [121,126] or fish [120,121,127] is associated with a 25–75% reduction in the ovarian cancer risk in comparison to the infrequent intake of poultry or fish. This may be because poultry contains a lower proportion of saturated fatty acids (>30% vs. ≥45%) and a higher proportion of polyunsaturated fatty acids (≥15% vs. <10%), in comparison to red meat [128].

In a large Danish population-based case-control study, the consumption of milk (200 mL/day), soured milk products (250 mL/day) and yoghurt (250 mL/day) was associated with a higher risk of developing ovarian cancer (ORs = 1.14, 1.49, and 1.65, respectively) [129]. In another population-based case-control study in the USA, an increased whole milk consumption and the intake of lactose was associated with a higher risk of developing ovarian cancer (highest quartile vs. lowest: OR = 1.97 *p* = 0.008). Using mouse models, our group evaluated whether different dietary protein qualities involving animal (casein: extracted from milk) or plant protein (soy protein) could inhibit ovarian cancer growth [130]. We found that the PI3K/AKT/mTOR pathway, a downstream signaling pathway of IGF-1, was more strongly activated by the intake of animal protein (casein) than the intake of plant protein (soy protein) in mice with ovarian cancer. Stated differently, a diet that was high in plant protein (soy protein) reduced the growth of human ovarian cancer cells in mice in comparison to a diet that was high in animal protein (casein), possibly through the relative inhibition of the IGF/Akt/mTOR pathway. 

### 4.3. Dietary Acrylamide 

No significant association was observed between the intake of acrylamide and the ovarian cancer risk in prospective cohort studies or a case-control study [131,132,133]. However, a prospective study within the Netherlands Cohort Study (NLCS) observed a statistically significant positive association between the consumption of large amounts of acrylamide and the overall ovarian cancer risk [134]. In addition, they analyzed acrylamide-gene interactions that were associated with the ovarian cancer risk [135], and concluded that there were statistically significant interactions between several single nucleotide polymorphisms (SNPs) in the *HSD3B1/B2* gene cluster and the acrylamide intake that were associated with the ovarian cancer risk, suggesting that acrylamide may cause the development of ovarian cancer through its effects on sex hormones [135]. In Japan, the dietary acrylamide intake was not associated with the ovarian cancer risk because the intake of acrylamide is lower in comparison to that in Western populations [136].

### 4.4. Phytoestrogens and Flavonoids 

Phytoestrogens are non-steroidal plant-derived compounds with a similar structure to endogenous estrogens. They are capable of showing both estrogen and antiestrogenic effects [137,138,139]. The main dietary phytoestrogens are isoflavones (mainly found in soy products), lignans and coumestans [140]. In addition to the estrogenic activity of isoflavones, which possibly led to the inhibition of the growth and proliferation of ovarian cell lines, isoflavones may regulate cancer inflammation pathways [141]. In a population-based case-control study, phytoestrogen was shown to reduce the ovarian cancer risk; however, the results did not reach statistical significance [139]. In a prospective population-based cohort study and two case-control studies, no association was found between the intake of phytoestrogens and the ovarian cancer risk [142,143]. When the analysis was limited to consumption of dietary isoflavones, however, isoflavones were found to have a protective effect against ovarian cancer in six studies (the RR ranging between 0.51 and 0.87) [139,143,144,145,146,147]. 

Flavonoids are a group of >4000 compounds, that are generally categorized into six major classes (flavonols, flavones, flavanones, flavanols, anthocyanidins and isoflavones) and a class of flavanol polymers [141]. Flavonoids have been reported to have antioxidant, antimutagenic and antiproliferative properties [148]. Plants contain bioactive constituents called flavonoids that modulate key cellular signaling pathways and which regulate multiple cancer-inflammation pathways and epigenetic cofactors [149]. A meta-analysis that included five cohort studies and seven case-control studies, suggested that the consumption of dietary flavonoids and subtypes (isoflavones, flavonols) had a protective effect against ovarian cancer and that—with the exception of flavone consumption—this consumption reduced the risk of developing ovarian cancer [147]. Similarly, another study showed that specific bioactive compounds, including flavonols and flavanones, which are present in plant-based foods, may reduce the ovarian cancer risk [149].

### 4.5. Calcium and Vitamin D 

In a population-based case-control study from the USA, the intake of calcium was reported to reduce the ovarian cancer risk (OR = 0.51, *p* = 0.009), but the intake of vitamin D was not [150]. Longer sun exposure in summer months was also found to be associated with a lower ovarian cancer risk (OR = 0.71, *p* = 0.049). The meta-analysis also supported the hypothesis that an increased calcium intake might reduce the ovarian cancer risk [151]. In a New England case-control study, a high total calcium intake was associated with a reduced overall ovarian cancer risk (quartile 4 [Q4 > 1319 mg/day] vs. quartile 1 [Q1 < 655 mg/day]; OR = 0.62) [152]. Histologically, the risk reduction was strongest for serous borderline and mucinous tumors. Similarly, in this study, a high total intake of vitamin D was not associated with the risk of ovarian cancer, but reduced the risk of developing serous borderline (Q4 > 559 IU/day, vs. Q1 < 164IU/day, OR = 0.51) and endometrioid tumors (Q4 vs. Q1, OR = 0.55). 

The calcium intake might down-regulate circulating parathyroid hormone (PTH) [151,153]. The reduction of PTH might reduce the hepatic and osteoblastic synthesis of insulin-like growth factor-1 (IGF-1) [153,154], which exerts an effect by increasing cellular proliferation and the inhibition of apoptosis [155]. The reduction of IGF-1 would weaken the mitogen effects on the pathogenesis of ovarian cancer [156,157]. 

Calcium may also protect against advanced ovarian cancer as a mediator of vitamin D-induced apoptosis [158]. Calcitriol, the active form of vitamin D, could suppress the growth of ovarian cancer in animal studies [159]. However, in most epidemiological studies, the intake of vitamin D was not associated with a reduced ovarian cancer risk [145,150,160,161,162], except for a possible association among overweight women [163]. In one cohort study, a higher serum 25(OH) D concentration at the time of the diagnosis of ovarian cancer was associated with longer survival [164]. The effect of vitamin D intake on ovarian cancer growth might be weaker in comparison to that of calcium.

### 4.6. Flaxseed 

Flaxseed is the richest vegetable source of ω-3 PUFAs. The chemo-preventive action of ω-3 PUFAs is their suppressive effect on the production of arachidonic acid-derived prostanoids, particularly prostaglandin E2 (PGE2) [165]. PGE2 is the most proinflammatory eicosanoid and one of the downstream products of two isoforms of cyclooxygenase (COX) enzymes: COX-1 and COX-2 [166] and is commonly elevated in different human cancers [167]. Using flaxseed-fed hens, Eilati et al. demonstrated that the flaxseed-mediated reduction in the severity of ovarian cancer in hens was correlated with the reduction in PGE2 in the ovaries of flaxseed-fed hens [168]. They also demonstrated that the lower levels of COX-2 and PGE2 were the main contributing factors in the chemo-suppressive role of long-term flaxseed consumption in ovarian cancer in laying hens [166].

## 5. Discussion

Table 2 summarizes the relationship between the dietary/nutritional intake and the risk of gynecologic cancers. The counterplots of diet seem to vary slightly for the three types of gynecologic cancers. We must continuously avoid the consumption of large amounts of high-risk diets and nutrients, and at the same time continuously consume preventive and reductive diets and nutrients. 

Regarding cervical cancer, etiology has been recognized. The most important roles of diet/nutrition in cancer are as prophylaxis and countermeasures against HPV infection. Cigarette smoking, which worsens all of the processes of cervical cancer is an important risk factor [11]. On the other hand, preventive and reductive factors include the intake of antioxidants (e.g., vitamin A, C, D and E, carotenoids, vegetables and fruits) [22,26,32,39]. These antioxidants may have different abilities to intervene in the natural history of diseases associated with HPV infection. Table 1 shows that the effects of vitamins C and E, calcium, carotenoids, vegetables and fruits differed among studies. In statistically analyzed human diet data, there may be a limit to our ability to observe the effects of a single nutrient, as many confounding factors influence the outcomes.

Regarding endometrial cancer, obesity should be avoided in peri-/post-menopause. The endometrial cancer risk can be explained by increased levels of peripheral estrogens as a result of the aromatization of androgens to estrogens in the adipose tissue of obese women [54]. Insulin resistance and metabolic syndrome are also thought to be risk factors [87,88]. Thus, we should take care to avoid continuously consuming fat energy and sugar, which seem to be the greatest dietary risk factors for endometrial cancer. 

The etiology of ovarian cancer has not been fully understood. Few risk factors have been suggested, including chronic inflammation, such as that associated with endometriosis and pelvic inflammatory diseases [104,105,106]. Thus, the long-term consumption of pro-inflammatory foods including saturated fat, carbohydrates and animal proteins, may have the potential to stimulate DNA damage, increased cell division of DNA repair aberrations, the promotion of angiogenesis, and to facilitate invasion [102,103]. Without a doubt, the intake of acrylamide is a risk factor for the development of both endometrial and ovarian cancer.

We should avoid the long-term consumption of large amounts of high-risk foods and nutrients. The key words for high-risk diets in gynecologic cancers seem to be “oxidant” “mutagenic” and “pro-inflammatory”. In contrast, the keywords for diets with preventive effects against gynecologic cancers seem to be “anti-oxidant” “anti-mutagenic” and “anti-inflammatory.” A common dietary component with a preventive effect against gynecologic cancers seems to be the soybean. The intake of soy protein inhibited the proliferative activities of certain cancers. In in vitro experiments, Rayaprolu et al. showed that soybean peptide fractions inhibited human blood (CCRF-CEM and Kasumi-3), breast (MCF-7) and prostate (PC-3) cancer cell lines by up to 68% [169]. Soy protein contains isoflavone (one of the flavonoids) that has been reported to have anti-oxidant, anti-mutagenic and anti-proliferative properties [148]. The long-term consumption of such good foods should be maintained for health. Western diets include the consumption of large amounts of animal protein, whereas traditional Japanese diets involve the consumption of large amounts of soy proteins. In Japan, the age-standardized ovarian cancer mortality rate is only 6.2 per 100,000 women [170]. In contrast, the age-standardized mortality rates in women in the USA and UK were 7.0 and 12.7 per 100,000 women, respectively [171,172]. 

Very recently, the dietary patterns have been reported to influence the course of several non-communicable diseases (NCDs), including cancer, through the modulation of gut microbiota. Klement and Pazienza reported that the Western diet, which is low in fiber and rich in sugar and processed foods, is also linked to a loss of microbial diversity, dysbiosis and a high-risk of obesity, cardiovascular disease, metabolic syndrome and cancer [173]. In short, a link between the diet, microbiota and cancer prevention and treatment has recently been unveiled, underscoring the importance of a new food culture based on limiting dietary surplus and a preference for healthier foods [173]. Exploring individuals’ microbial profiles will prove useful for establishing personalized strategies of microbiota manipulation in order to improve the cancer therapeutic outcome [174]. 

## 6. Conclusions

The findings from this review suggest that the long-term consumption of large amounts of high-risk foods and nutrients should be avoided in order to reduce the risk of gynecologic cancers. In addition, the long-term consumption of foods shown to be preventive against each cancer should conversely be promoted in order to maintain good health. 

In order to reduce the cervical cancer risk, we may need to promote the long-term consumption of antioxidants, such as vitamins A, C, D and E, carotenoids, vegetables and fruits, and avoid cigarette smoking. To reduce the endometrial cancer risk, we should take care to avoid continuously consuming fat, energy sugar and acrylamide. To reduce the ovarian cancer risk, we should take care to avoid the long-term consumption of pro-inflammatory foods, including saturated fat, carbohydrates, animal proteins and acrylamide.

The present study reviewed the relationship between the dietary and nutrient intake and the gynecologic cancer risk. Most papers were epidemiological studies and clinical trials. There were few experimental studies. Thus, the mechanisms through which diet and nutrition influence the development of gynecologic cancers are not clearly understood. Further research using in vitro and in vivo approaches is needed to clarify these mechanisms. 

## Figures and Tables

**Table 1 healthcare-07-00088-t001:** The preventive and reductive effects of the dietary/nutritional intake on human papillomavirus (HPV) infection, CIN 1, CIN 2, CIN 3 and cervical cancer.

Cervical Disease	Preventive and Reductive Diets/Nutrients						
HPV infection	Mediterranean diet [20],	papaya	vitamin-C [24],	vegetables [22,23],	carotenoids [23,24],	fruits [20]	
CIN 1 (SIL)		papaya [25],			carotenoids (β-cryptoxanthin, α-carotene) [25]		
	vitamin D [26,27], vitamin A (retinol) [18,28]						
CIN 2	nuts, legumes [29],			onion vegetables [29]			
CIN 2/3				vegetables [30,31],		fruits [30],	green tea [31]
	multivitamins,	vitamin C [32], vitamin E [32],					calcium +/− [32,33]
	folate [34,35]						
CIN 3			vitamin E [22],	vegetables [37],	carotenoids [22],	fruits [37],	lycopene [22,37]
Cervical Cancer					carotenoids [39,40,41],	fruits [42,43],	sulforaphane [45] *
					tocopherol [41], tea polyphenols [46] *		
					green tea [47] *, flavonoids * (quercetin [48,50] *		
					apigenin [49] *, genistein [51] *)		
					polyunsaturated fatty acid [52]		

[ ] = reference number. * = experimental study. Others are clinical data. Nutrients with underline come out several times.

**Table 2 healthcare-07-00088-t002:** The relationship between the dietary/nutritional intake and the risk of gynecologic cancers.

Gynecologic Cancer	High-Risk Diets/Nutrients	Preventive and Reductive Diets/Nutrients
Cervical Cancer	cigarette	multi vitamin, vitamin A, vitamin C, vitamin D, vitamin E, papaya
	Western diet, oleic acid *	Mediterranean diet, carotenoids, fruits, vegetables, legume,
		lycopene, green tea *, folate, sulforaphane *, polyphenol *
		Flavonoids *, polyunsaturated fatty acid *, calcium (+/−)
Endometrial Cancer	fat energy, sugar, polyunsaturated fatty acid,	alcohol, proanthocyanidin, carbohydrate (+/−)
	docosahexaenoic acid, saturated fat,	Kaempherol *, isoflavone
	Adiponectin *, acrylamide *, alcohol	
Ovarian Cancer	saturated fat, carbohydrate, trans-fat,	poly-unsaturated fatty acid, ω-3 fatty acid,
	red meat, processed meat,	poultry, fish, soy protein *,
	milk (casein) *, acrylamide *	flavonoids (isoflavone *, flavonol, flavanone)
		Calcium *, vitamin D *, flaxseed *

* = experimental study. Others are clinical data.

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
