# Peer review of "The Effects of the Dietary and Nutrient Intake on Gynecologic Cancers"

_healthcare, 2019, doi:10.3390/healthcare7030088_

Round 1

Reviewer 1 Report

In the current review article, Dr. Koshiyama has reviewed the literature indicating the role of diets and nutrients on the emergence of three different types of gynecologic cancers. Content of the manuscript is great but storytelling is not appealing. I found that the majority of sentences and sections starts abruptly throughout the manuscript, which make it hard to follow and boring. The introductory part is missing in almost all sections. In the Introduction section Dr. Koshiyama claims that diet is responsible for the higher rate of cancer in western countries. I advise not to use such a strong claim because effects of other factors such as living style and environmental factors cannot be ignored. I suggest that the Author should revise the manuscript, along with the abstract and resubmit. Wherever needed following flow can be adopted: introduction (background), statement of problem, methodology (literature review), analysis, conclusions and recommendations for further research.

Author Response

<Answer to the Referee 1>

Corrected parts are described with red color.

Comments and Suggestions for Authors

In the current review article, Dr. Koshiyama has reviewed the literature indicating the role of diets and nutrients on the emergence of three different types of gynecologic cancers. Content of the manuscript is great but storytelling is not appealing. I found that the majority of sentences and sections starts abruptly throughout the manuscript, which make it hard to follow and boring.

Yes. Thank you for good advices. I inserted the following sentences:

I inserted following sentences in Introduction:This paper reviews the current issues (a review of the relevant literature), and the effects of the dietary and nutrient intake on three types of gynecologic cancer (cervical, endometrial and ovarian cancers) and discusses the roles of the dietary and nutrient intake in relation to each type of cancer.

I inserted following sentences in 2. Cervical Cancer: Cervical cancer develops through HPV infection, CIN1, CIN2 and CIN3. To provide an understanding of the preventive and complementary effects, the relationship between the effects of the dietary and nutrient intake, and the development of cervical cancer (HPV infection, CIN1, CIN2, CIN3 and cervical cancer) are now described.

I inserted following sentences in 3. Endometrial Cancer: In the case of endometrial cancer, the direct etiology has not been clearly understood. To provide an understanding of high-risk diets and the preventive effects of diet, the relationship between dietary intake and the endometrial cancer risk is discussed.

I inserted following sentences in 4. Ovarian Cancer: The etiology of ovarian cancer is not understood. To provide an understanding of high-risk diets and the preventive effects of diet, the relationship between dietary intake and the ovarian cancer risk is discussed.

The introductory part is missing in almost all sections. In the Introduction section Dr. Koshiyama claims that diet is responsible for the higher rate of cancer in western countries. I advise not to use such a strong claim because effects of other factors such as living style and environmental factors cannot be ignored.

Yes. Thank you for good advices. I corrected Introduction. I changed the previous sentences to the following sentences: Among them, the contribution of diet to the cancer risk in developed countries has been considered to be relatively higher [3-5], whereas that in developing countries has been considered to be lower, perhaps around 20% [6,7].

I suggest that the Author should revise the manuscript, along with the abstract and resubmit. Wherever needed following flow can be adopted: introduction (background), statement of problem, methodology (literature review), analysis, conclusions and recommendations for further research.

Yes. Thank you for good advices. I erased the following sentences in 2.Cervical Cancer: Regarding the preventive and complementary effects of diet in relation to cervical cancer, most papers were epidemiological studies and clinical trials that aimed to further elucidate the role of diet in the management of CINs and cervical cancers in women; there have been few experimental studies. Thus, the mechanisms underlying the effects of diet and nutrition have not been clearly understood.

And I also erased the following sentences in 5. Discussion: I reviewed the relationship between the dietary and nutrient intake and the gynecologic cancer risk. Most papers were epidemiological studies and clinical trials. There were few experimental studies. Thus, the mechanisms through which diet and nutrition influence the development of gynecologic cancers are not clearly understood. Further research using in vitro and in vivo approaches is needed to clarify these mechanisms.

And I inserted the following sentences in 6. Conclusion. The present study reviewed the relationship between the dietary and nutrient intake and the gynecologic cancer risk. Most papers were epidemiological studies and clinical trials. There were few experimental studies. Thus, the mechanisms through which diet and nutrition influence the development of gynecologic cancers are not clearly understood. Further research using in vitro and in vivo approaches is needed to clarify these mechanisms.

To provide an understanding of the relationship between dietary intake and the CIN risk, I deleted the paragraph of 2.5 CIN1, CIN2,CIN3 and Nutrition

To provide an understanding of the relationship between dietary intake and the CIN/cervical cancer risk, I changed Table 1 and Table 2.

Reviewer 2 Report

The review titled “The effects of the dietary and nutrient intake on gynecologic cancers” by Masafumi Koshiyama presents some interesting points on the role of nutrition on the three types of gynecological cancers.

Comments:

Experimentally it is difficult to study dietary factors in causing and preventing cancer due to the presence of several confounding factors and therefore papers contradicting each other are not uncommon. The author has presented more than one reference in several instances throughout the review. However, care should be taken that this is done uniformly for all nutrients presented, wherever possible present a balanced view on the subject matter in the context of recent papers and consider adding multiple references to ensure that the point made has had solid experimental evidence from at least two independent groups. As examples-

#Lines 66-77 present two references supporting intake of Mediterranean diet-although the first reference is for prevention of infection and the second one is for persistence of infection. Are there any more references to support this? In lines 75-77 there is single reference for dietary intake of carotenoids and its association with reduction of HPV incidence. This reference is more than a decade old. Has there been more work in this field? If not, present in a couple lines why this paper is worth mentioning-was the size of the population big enough or the results really strong enough?

#is there only a single study [referenced as 24] showing positive effect of Vit D on CIN1 patients? Is there any study presenting evidence to the contrary or supporting the claims made in 24?

The point that I am trying to make here is that in view of limited space the author should provide evidence for convincing studies for one [or two dietary factors] providing ample support for the point made.

Author Response

<Answer to the Referee 2>

Corrected parts are described with red color.

Comments and Suggestions for Authors  

Comments:

Experimentally it is difficult to study dietary factors in causing and preventing cancer due to the presence of several confounding factors and therefore papers contradicting each other are not uncommon. The author has presented more than one reference in several instances throughout the review. However, care should be taken that this is done uniformly for all nutrients presented, wherever possible present a balanced view on the subject matter in the context of recent papers and consider adding multiple references to ensure that the point made has had solid experimental evidence from at least two independent groups. As examples-

#Lines 66-77 present two references supporting intake of Mediterranean diet-although the first reference is for prevention of infection and the second one is for persistence of infection. Are there any more references to support this?

Yes. Thank you for good advices. I added the following sentence and references in 2.1. HPV Infection and Nutrients: A Western diet has been reported to lead to increased inflammation, reduced infection control and increased risk of developing auto-inflammatory disease [21].

Supporting these results, Sedjo et al. reported that higher levels of vegetable consumption were associated with a 54% reduction in the risk of HPV persistence (OR: 0.46) in a prospective cohort study [23] .

 In lines 75-77 there is single reference for dietary intake of carotenoids and its association with reduction of HPV incidence. This reference is more than a decade old. Has there been more work in this field? If not, present in a couple lines why this paper is worth mentioning-was the size of the population big enough or the results really strong enough?

Yes. Thank you for good advices. I added the following sentence and

references in 2.1. HPV Infection and Nutrients: The intake of carotenoid (lutein) was also reported to be associated with a 50-63% reduction in the risk of HPV persistence in a prospective cohort study [23].

And I added the following sentence and references in 2.2. CIN 1 and Nutrients: These data are consistent with data from the report of Giuliano et al. [24], which showed a reduced incidence of type-specific HPV persistence.

#is there only a single study [referenced as 24] showing positive effect of Vit D on CIN1 patients? Is there any study presenting evidence to the contrary or supporting the claims made in 24?

Yes. Thank you for good advices. I added the following sentence and

references in 2.2. CIN 1 and Nutrients:  The report of Schulte-Uebbing et al. which showed that treatment with vitamin D vaginal suppositories (12,500 IU, three nights a week, for 6 weeks) resulted in antidysplastic effects in patients with CIN 1, but dnot those with CIN 2, supports this hypothesis [27].

The point that I am trying to make here is that in view of limited space the author should provide evidence for convincing studies for one [or two dietary factors] providing ample support for the point made.

Yes. Thank you for good advices. I added the following sentences in 2.5. Cervical Cancer and Nutrients: Another case-control study supported an inverse association between ISC and fresh fruit intake [43].

I added the following sentences in 3.7. Flavonoid:  Another report indicated that the intake of soy-containing foods was associated with a lower endometrial cancer risk [98].

I added the following sentences in 4.1. Pro-inflammatory Food:  It was indicated that polyunsaturated fatty acids, such as DHA inhibited proliferation in ovarian cancer cell lines via G1 cell cycle arrest and the induction of apoptosis and cellular stress [118].

I also changed Table 1.

Round 2

Reviewer 1 Report

Dr. Koshiyama have made satisfactory changes in the manuscript. Please align the contents of table-1. I do not have any additional concerns.

This manuscript is a resubmission of an earlier submission. The following is a list of the peer review reports and author responses from that submission.

Round 1

Reviewer 1 Report

The author has summarized the clinical and epidemiological evidences to support the role of dietary and nutrient intake on gynecological cancers such as cervical, endometrial and ovarian. In this review, the authors discussed several interesting dietary components and their effect on gynecological cancers. This review has shed light on some interesting relationship between food and cancer and indicated the requirement of further research on the mechanism of this relationship. Some of the inquiries below are awaiting for response.

1. Three major types of cancer were discussed in the manuscript. For endometrial and ovarian cancer, the perspectives of descriptions were from the nutrients themselves with detailed discussion. For cervical cancer, the perspectives were from the cancer types of CIN1 through CIN3. Is it possible to provide more involved nutrients to cervical cancer compared to other two types of cancer, which will strengthen the academic significance of this review article?

2. In line 143 to 146, the sentence was incomplete.

3. In reference section, some of the references were not correctly formatted. The errors of capitalization and space things were shown.

Author Response

Answers to the reviewers.

 I corrected the sentences with red color.

1. Comments and Suggestions for Authors (Referee 1) & Answers

1. Three major types of cancer were discussed in the manuscript. For endometrial and ovarian cancer, the perspectives of descriptions were from the nutrients themselves with detailed discussion. For cervical cancer, the perspectives were from the cancer types of CIN1 through CIN3. Is it possible to provide more involved nutrients to cervical cancer compared to other two types of cancer, which will strengthen the academic significance of this review article?

 èThank you very much for good advice. The case of cervical cancer is based on HPV infection unlike endometrial and ovarian cancers. The antioxidants, such as vitamin A, C, D and E, vegetables and fruits have abilities of reductive HPV infection. But these antioxidants have different abilities to intervene the natural history of cervical cancer development. Thus, I had better describe these effects in accordance with its stage.

2. In line 143 to 146, the sentence was incomplete.

è Yes, I corrected the writing before that in the following way.

in the daily total vegetable intake (HR 0.85) was observed [38].

3. In reference section, some of the references were not correctly formatted. The errors of capitalization and space things were shown.

èThank you very much for your advice. I carefully corrected references. I showed the corrected parts with red color!

Reviewer 2 Report

This is an updated review dealing with diet and gynecologic cancer.

It is well organized and well structured.

Although I agree with the author that the mechanisms through which diet and nutrition influence the development of gynecologic cancers are not clearly understood, we already know that diet influences the course of several Non Communicable Disease (NCDs) including cancer through the modulation of gut microbiota.

Since I understand that the topic of microbiota was behind the scope of this review I strongly suggest to add at least in the discussion section few sentences about the positive effect of certain diets on microbiota and cancer treatmens referring the following articles:

1)  Impact of Different Types of Diet on Gut Microbiota Profiles and Cancer Prevention and Treatment. Klement RJ, Pazienza V. Medicina (Kaunas). 2019 Mar 29;55(4).

2) Pharmacomicrobiomics: exploiting the drug-microbiota interactions in anticancer therapies. Panebianco C, Andriulli A, Pazienza VMicrobiome. 2018 May 22;6(1):92. d

Author Response

Answers to the reviewers.

 I corrected the sentences with red color.

 Comments and Suggestions for Authors (Referee 2) & Answers

 This is an updated review dealing with diet and gynecologic cancer.

It is well organized and well structured.

Although I agree with the author that the mechanisms through which diet and nutrition influence the development of gynecologic cancers are not clearly understood, we already know that diet influences the course of several Non Communicable Disease (NCDs) including cancer through the modulation of gut microbiota.

Since I understand that the topic of microbiota was behind the scope of this review I strongly suggest to add at least in the discussion section few sentences about the positive effect of certain diets on microbiota and cancer treatmens referring the following articles:

1)  Impact of Different Types of Diet on Gut Microbiota Profiles and Cancer Prevention and Treatment. Klement RJ, Pazienza V. Medicina (Kaunas). 2019 Mar 29;55(4).

2) Pharmacomicrobiomics: exploiting the drug-microbiota interactions in anticancer therapies. Panebianco C, Andriulli A, Pazienza V. Microbiome. 2018 May 22;6(1):92. d

èThank you very much for your good ideas. I cited two papers and added the following paragraph in Discussion.

Very recently, the dietary patterns have been reported to influence the course of several non-communicable diseases (NCDs), including cancer, through the modulation of gut microbiota. Klement and Pazienza reported that the Western diet, which is low in fiber and rich in sugar and processed foods, is also linked to a loss of microbial diversity, dysbiosis and a high risk of obesity, cardiovascular disease, metabolic syndrome and cancer [173]. In short, a link between the diet, microbiota and cancer prevention and treatment has recently been unveiled, underscoring the importance of a new food culture based on limiting dietary surplus and a preference for healthier foods [173]. Exploring individuals’ microbial profiles will prove useful for establishing personalized strategies of microbiota manipulation in order to improve the cancer therapeutic outcome [174].

Reviewer 3 Report

The topic is likely to be of interest to the reader of Healthcare, however, the manuscript requires extensive revisions.

General comments:

The manuscript needs to be rewritten into appropriate paragraphs. Please see the following url for guidance if necessary: https://www.wikihow.com/Write-a-Paragraph

At present many of the paragraphs contain numerous ideas that you have not related to each other. This leads to a manuscript that is disjointed and disorganised, as well as statements that are not justified.  

The manuscript could be improved by the inclusion of tables and figures. For example, section 2 on cervical cancer, rather than containing numerous subheadings with a mix of clinical trials/human studies, animal models and in vitro experiments, sometimes in one paragraph, sub-headings 2.2-2.5 could be presented in a table. This table would include the citations, study design, findings etc. It would make sense to include other tables where relevant.

In addition, when results from different studies are not consistent with one another, you need to discuss this discrepancy, and not ignore it, as you have done.

When referring to a particular study and the risk of cancer, one assumes you are not referring to the three gynaecologic cancers you have focused on. If this is the case you need to relate the findings you present, with your topic e.g. lines 216-219.

The above comments need to be applied throughout the manuscript.

Specific comments:

Abstract:

Line 8 -"higher" is a comparative term - higher than what?

Methods? Refer to the typed of studies you discuss.

Lines 14-15 - this is an abrupt change. Please improve the flow.

Remain focused on dietary and nutrient intake, seeing that this is the title you have selected. 

Line 22 - which mechanisms are you referring to.

Introduction:

Estimates change, and hence it is important that you update the introduction to include more recent estimates.

Line 32-33 - be specific, you are focusing on three types of gynaecological cancer, not cancer in general. Also, why is it important "for public health"

References:

These appear to have gone awry after reference 7 in the intext citations. Numerous errors are present in the reference list. (Due to the presence of spelling errors, I can only assume that you have not used a reference management system, or entered the references automatically. This makes correct referencing very difficult indeed. I suggest you download free software to assist you with this, or use that provided by your University).

Line 38 - you have included far too many references, some of which are irrelevant or at least, unnecessary.

Include a reference at the end of line 40, end of the sentences in lines 135,184, 186, 202, 227, amongst others.

If you refer to "studies" please include more than one citation.

Section 2 - explain CIN1-3, also, E6 and E7.

Line 199 - correct this sentence. You refer to "either", yet there only appears to be one group?

Section 3.2. The first couple of sentences could be included in the introduction to section 3 (avoid repetition).

Section 3.3, first paragraph, first sentence is incomplete; in the last sentence of this paragraph, relate the findings to the other studies you have discussed in this paragraph and be consistent with the terms you use.

I think you get the idea of the type of correction you need to make throughout the manuscript.

Author Response

Answers to the reviewers.

 I corrected the sentences with red color.Comments and Suggestions for Authors (Referee 3) & Answers

The topic is likely to be of interest to the reader of Healthcare, however, the manuscript requires extensive revisions.

*General comments:

The manuscript needs to be rewritten into appropriate paragraphs. Please see the following url for guidance if necessary: https://www.wikihow.com/Write-a-ParagraphAt present many of the paragraphs contain numerous ideas that you have not related to each other. This leads to a manuscript that is disjointed and disorganised, as well as statements that are not justified.  

èThank you very much for your important advice. But I wrote this review article in accordance with the form of a previous published review article in “Healthcare.” I am sorry that I can not change this paragraph form drastically. Regarding cervical cancer, the effects of diets and nutrition are very complex. Therefore, I inserted the following sentences in 2.6.

Table 1 summarizes the preventive and reductive effects of the dietary/nutritional intake on HPV infection, CIN 1, CIN 2, CIN 3 and cervical cancer. In Table 1, the main preventive and reductive factors are antioxidants, such as vitamin A, C, D and E, carotenoids, vegetables and fruits. These antioxidants may have different intervention effects on the natural history of diseases associated with HPV infection. Vitamin A, C, D and E show the preventive and reductive effects on CIN, whereas carotenoids, fruits, sulforaphane, polyphenol and flavonoids (including quercetin, apigenin and genistein) show reductive effects on cervical cancer.

And I added the following 5. Conclusion.

5. Conclusions

 The findings from this review suggest that the long-term consumption of large amounts of high-risk foods and nutrients should be avoided in order to reduce the risk of gynecologic cancers. In addition, the long-term consumption of foods shown to be preventive against each cancer should conversely be promoted in order to maintain good health.

  In order to reduce the cervical cancer risk, we may need to promote the long-term consumption of antioxidants, such as vitamin A, C, D and E, carotenoids, vegetables and fruits, and avoid cigarette smoking. To reduce the endometrial cancer risk, we should take care to avoid continuously consuming fat, energy sugar and acrylamide. To reduce the ovarian cancer risk, we should take care to avoid the long-term consumption of pro-inflammatory foods, including saturated fat, carbohydrates, animal proteins and acrylamide.

*The manuscript could be improved by the inclusion of tables and figures. For example, section 2 on cervical cancer, rather than containing numerous subheadings with a mix of clinical trials/human studies, animal models and in vitro experiments, sometimes in one paragraph, sub-headings 2.2-2.5 could be presented in a table. This table would include the citations, study design, findings etc. It would make sense to include other tables where relevant.

èThank you very much for your kind advices. Yes, I added Table 1. In Table 1, the effects of diets and nutrition are divided into HPV infection, CIN 1, CIN 2, CIN 3 and cervical cancer. And I added reference-numbers and experimental study stamps.

*In addition, when results from different studies are not consistent with one another, you need to discuss this discrepancy, and not ignore it, as you have done.

èThank you very much for an important advice. Yes, I added this point in Discussion. In Table 1, the effects of vitamins A, C, D and E were found to differ among studies, showing no consistency. In statistically analyzed human diet data, there may be a limit to our ability to observe the effects of a single nutrient, as many confounding factors influence the outcomes.

*When referring to a particular study and the risk of cancer, one assumes you are not referring to the three gynaecologic cancers you have focused on. If this is the case you need to relate the findings you present, with your topic e.g. lines 216-219.

èThank you for your advice. In order to avoid misunderstanding, I corrected the writing before that in the following way. I inserted “endometrial”.Among women with waist –to–hip ratio (WHR; a marker of central obesity) of0.85, the endometrial cancer risk of the women in the highest tertile of added sugar intake was significantly higher than that of the women in the lowest tertile of added sugar intake (OR = 2.50).

The above comments need to be applied throughout the manuscript.

Specific comments:

Abstract:

*Line 8 -"higher" is a comparative term - higher than what?

èYes. Thank you very much for your advice. I corrected the sentence.

“to be higher in advanced countries than in developing countries.”

*Methods? Refer to the typed of studies you discuss.

èThank you for your advice. I inserted (a review of the literature) in Abstract.

*Lines 14-15 - this is an abrupt change. Please improve the flow.

èThank you for your advice. I inserted the following sentences:          For endometrial cancer, in Abstract.

Remain focused on dietary and nutrient intake, seeing that this is the title you have selected. 

èYes, it is difficult to show all food effects briefly.  I had intended to show it using the next sentences: “The main preventive and reductive factors of cervical cancer are antioxidants, such as vitamin A, C, D and E, carotenoids, vegetables and fruits.”  “Thus, we must mainly take care to avoid the continuous intake of fat energy and sugar (regarding endometrial cancer risk).” “To the best of our knowledge, the long-term consumption of pro-inflammatory foods, including saturated fat, carbohydrates and animal proteins is a risk factor (for ovarian cancer).”

*Line 22 - which mechanisms are you referring to.

èThank you very much for your advice. I corrected the writing before that in the following way. “Thus, further research using in vitro and in vivo approaches is needed to clarify the effects of the dietary and nutrient intake in detail.”

Introduction:

*Estimates change, and hence it is important that you update the introduction to include more recent estimates.

èThank you for advice. I searched new more recent papers. But I can not find them. I am sorry!

*Line 32-33 - be specific, you are focusing on three types of gynaecological cancer, not cancer in general. Also, why is it important "for public health"

èGenerally, understanding the effects of diet and nutrition on various cancer risks is important for health. But, I focused on gynecological cancers this time for Special Issues.

References:

These appear to have gone awry after reference 7 in the intext citations. Numerous errors are present in the reference list. (Due to the presence of spelling errors, I can only assume that you have not used a reference management system, or entered the references automatically. This makes correct referencing very difficult indeed. I suggest you download free software to assist you with this, or use that provided by your University).

èThank you very much for your very kind advice. But I do not have searching soft yet. Then, I carefully corrected references. The corrected parts are shown with red color.

*Line 38 - you have included far too many references, some of which are irrelevant or at least, unnecessary.

Include a reference at the end of line 40, end of the sentences in lines 135,184, 186, 202, 227, amongst others.

è That is a misunderstanding. I wrote no references. On line 38, numbers of 16, 18, 31,,,,,,,,,,68 are HPV types.

*If you refer to "studies" please include more than one citation.

èThank you. The "studies" is shown in a meta-analysis.Chu et al. performed a meta-analysis of 17 studies to investigate the effect of the overall energy intake on the endometrial cancer risk [55].” Therefore, I only cited reference 55.

*Section 2 - explain CIN1-3, also, E6 and E7.

èThank you. The words of CIN1-3, E6 and E7 are common knowledge for gynecologists and oncologists. Cervical cancer develops via CIN1, CIN2 and CIN3.  E6 and E7 are the malignant protein of HPV.

*Line 199 - correct this sentence. You refer to "either", yet there only appears to be one group?

èThank you for your advice. Chu et al. performed a meta-analysis of 17 studies (groupe).

*Section 3.2. The first couple of sentences could be included in the introduction to section 3 (avoid repetition).

èThank you very much for a kind advice. Yes, I erased two sentences (online 206- 209).

*Section 3.3, first paragraph, first sentence is incomplete; in the last sentence of this paragraph, relate the findings to the other studies you have discussed in this paragraph and be consistent with the terms you use.

èThank you very much for your kind advice. Yes, I corrected the writing before that in the following way. “with endometrial cancer and 908 controls. The intake of” èwith endometrial cancer and 908 controls, the intake of

I think you get the idea of the type of correction you need to make throughout the manuscript.

Thank you very much for your detailed advices!!

Round 2

Reviewer 3 Report

Manuscripts need to be well written in order for the Reader (whose first language may or may not be English) to readily understand the content upon reading it for the first time. In order for this to occur, the manuscript needs to be well structured, and flow from one paragraph to the other, and one sentence to the next. To do otherwise is to disrespect the Reader. To provide an explanation that it isn't necessary to restructure paragraphs such that they focus on one idea, rather than multiple study types and numerous nutrients (in vitro, in vivo) etc., simply because you have previously presented a manuscript in this way, is not acceptable.

Readers with an interest in particular nutrients and/or gynaecological cancers need to understand what has been written, and it is therefore necessary to write accordingly.

In addition, the literature needs to be critically analysed rather than merely presented.